# AGROFLUX: A SPATIAL-TEMPORAL BENCHMARK FOR CARBON AND NITROGEN FLUX PREDICTION IN AGRICULTURAL ECOSYSTEMS

## ABSTRACT

Agroecosystem, which heavily influenced by human actions and accounts for a quarter of global greenhouse gas emissions (GHGs), plays a crucial role in mitigating global climate change and securing environmental sustainability. However, we can't manage what we can't measure. Accurately quantifying the pools and fluxes in the carbon, nutrient, and water nexus of the agroecosystem is therefore essential for understanding the underlying drivers of GHG and developing effective mitigation strategies. Conventional approaches like soil sampling, process-based models, and black-box machine learning models are facing challenges such as data sparsity, high spatiotemporal heterogeneity, and complex subsurface biogeochemical and physical processes. Developing new trustworthy approaches such as AI-empowered models, will require the AI-ready benchmark dataset and outlined protocols, which unfortunately do not exist. In this work, we introduce a first-of-its-kind spatial-temporal agroecosystem GHG benchmark dataset that integrates physics-based model simulations from Ecosys and DayCent with real-world observations from eddy covariance flux towers and controlled-environment facilities. We evaluate the performance of various sequential deep learning models on carbon and nitrogen flux prediction, including LSTM-based models, temporal CNN-based model, and Transformer-based models. Furthermore, we explored transfer learning to leverage simulated data to improve the generalization of deep learning models on real-world observations. Our benchmark dataset and evaluation framework contribute to the development of more accurate and scalable AI-driven agroecosystem models, advancing our understanding of ecosystem-climate interactions.

## 1 INTRODUCTION

Agricultural ecosystems play a critical role in global carbon and nitrogen cycles as they significantly influence climate change through the exchange of greenhouse gases like carbon dioxide ($CO_2$) and nitrous oxide ($N_2O$) with the atmosphere (Shukla et al., 2022; Clark et al., 2020). Gross Primary Productivity (GPP) and $CO_2$ fluxes represent the carbon uptake through photosynthesis and the release through metabolic processes. $N_2O$, with a global warming potential approximately 300 times that of $CO_2$, is primarily produced during soil nitrogen transformations through microbial processes like nitrification and denitrification, driven by agricultural fertilization management (Griffis et al., 2017). Accurately modeling these fluxes is therefore critical for understanding their spatial and temporal dynamics and developing effective climate change mitigation strategies.

Conventionally, process-based models (PBMs) have been widely used for estimating agricultural carbon and nitrogen fluxes. These models use mathematical equations to explain the relationships between environmental variables and biogeochemical processes. For example, the Ecosys model (Grant, 2001) simulates carbon uptake, respiration, and allocation processes influenced by soil temperature, moisture, and nutrient availability, while DayCent (Del Grosso et al., 2001; 2005) focuses on daily time-step simulations of carbon and nitrogen dynamics in various ecosystems. These PBMs involve complex parameterization that incorporate mechanistic relationships and physical laws to predict desired output variables under specified environmental and management scenarios, including those outside the range of historical observations. However, due to their extensive

parameterization and complex calculations, these models are often structurally biased and computationally expensive, which limit their applicability in real-time and large-scale scenarios.

Recently, there has been a growing interest in using machine learning methods for agricultural flux estimation (Reichstein et al., 2019; Xu & Valocchi, 2015; O'Gorman & Dwyer, 2018). Compared to traditional PBMs, machine learning models provide a more computationally efficient approach that can effectively capture complex nonlinear patterns in flux data. However, due to the limited availability of large observation datasets, traditional machine learning approaches often lack generalizability to out-of-sample prediction scenarios, e.g., unseen time periods or spatial regions. This challenge is further exacerbated by the high spatial and temporal variability in agricultural flux observations. Existing agricultural benchmark datasets focus predominantly on satellite-based estimates of crop yield, crop type, and field boundaries (Kerner et al., 2025; Paudel et al., 2025; Sykas et al., 2021). However, they omit critical biogeophysical and biogeochemical variables of carbon–nitrogen-water-thermal cycling, and therefore cannot support accurate quantification of greenhouse gas (GHG) fluxes.

In this paper, we introduce AgroFlux, which is a benchmark suite for agricultural flux prediction. AgroFlux defines standardized scenarios—temporal extrapolation, spatial extrapolation, standardized prediction tasks—predicting simulated data, predicting observation data, and transfer learning, and specifies consistent evaluation metrics—$R^2$, RMSE, MAE for fair comparison of models and transfer learning techniques. The underlying dataset integrates physical simulations generated by Ecosys and DayCent with true observations at a daily scale, forming the foundation of these tasks. The simulated data contain flux predictions and variables representing underlying biogeophysical and chemical processes, across numerous sites and different management practices, providing a comprehensive resource for evaluating model performance under controlled conditions. The observational datasets, consisting of measured fluxes from multiple controlled-environment facilities over long time periods, represent real-world conditions with varying environmental factors. Additionally, our underlying dataset incorporates a wide range of driver variables collected from different sources, which account for both temporal dynamics (e.g., weather changes) and spatial variability (e.g., soil properties and management practices). Together, these components establish AgroFlux as a benchmark, enabling researchers to evaluate new models, transfer learning techniques, and domain-knowledge integration under unified protocols. We also provide baseline performance of six machine learning models (LSTM, EA-LSTM, TCN, Transformer, iTransformer, Pyraformer) and two transfer learning techniques (pretrain-finetune, and adversarial training) across all benchmark tasks. These results serve as reference points, positioning AgroFlux as a leaderboard benchmark for agricultural prediction.

## 2 RELATED WORKS

While several benchmark datasets exist for agricultural applications, they primarily focus on crop yield prediction, land cover classification, or specific environmental variables. For example, the Crop Yield Prediction Dataset (Khaki et al., 2020) provides historical yield data along with weather and soil information but lacks comprehensive biogeochemical variables. The FLUXNET2015 dataset (Pastorello et al., 2020) offers valuable flux measurements but has limited coverage of agricultural sites and lacks detailed management practice information.

For GHG fluxes, AmeriFlux and FLUXNET networks have compiled eddy covariance measurements across diverse ecosystems (Novick et al., 2018), but agricultural sites remain underrepresented. The Global $N_2O$ Database (de Klein et al., 2020) collects $N_2O$ emission measurements from agricultural fields globally, but lacks the temporal resolution needed for process-level understanding. Similarly, the GHG-Europe database (Kutsch et al., 2010) provides flux measurements from European croplands but with limited spatial coverage and variable temporal resolution. The X-BASE (Nelson et al., 2024) and its origin FLUXCOM (Jung et al., 2020), which upscale the flux measurements from FLUXNET by utilizing satellite remote sensing, weather drivers, and machine learning ensembles, can potentially benchmark the PBMs and ML models, but they are limited in agroecosystem applications due to a lack of management information and key variables representing the nitrogen cycles, and biogeophysical and chemical processes.

In summary, most existing agricultural benchmarks focus on certain aspects of agricultural systems, such as yield prediction or land cover classification. These datasets often lack the integration

of comprehensive biogeochemical variables necessary for accurate greenhouse gas flux prediction. Our AgroFlux benchmark addresses these limitations by providing an integrated dataset that combines simulated data from process-based models with real-world observations across multiple locations and management practices, offering a comprehensive resource for evaluating machine learning models on agricultural carbon and nitrogen flux prediction tasks.

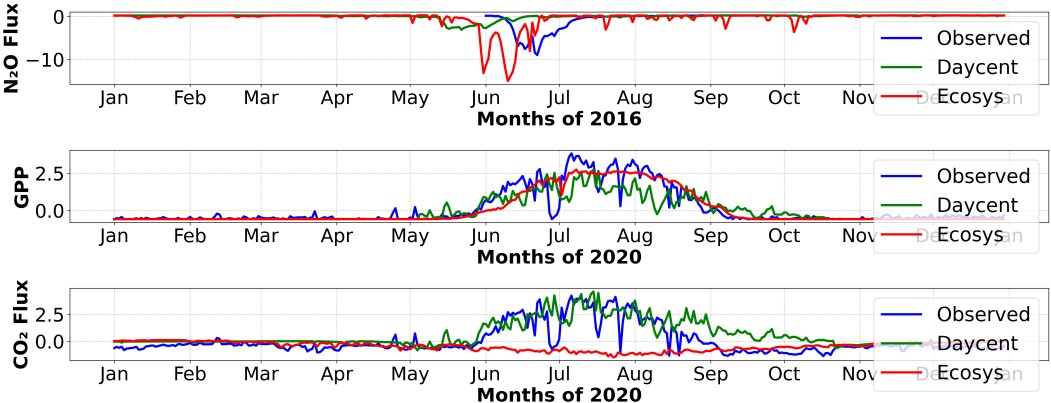

Figure 1: Temporal variations in $N_2O$ flux (top), GPP (middle), and $CO_2$ flux (bottom) from both simulated and observation sets. The unit for all three variables is (g C m$^{-2}$ day$^{-1}$).

## 3    AGROFLUX: DATASET CONSTRUCTION

The dataset used by AgroFlux integrates both simulated and observational data of key variables in agricultural ecosystems. The simulated data are generated by PBMs and are designed to simulate realistic agroecosystem dynamics by capturing the complex interactions between various environmental, soil, and management factors. The observation data are collected for the measurement of agricultural carbon and nitrogen outcomes. In particular, we focus our evaluation on three key variables: GPP, $CO_2$, and $N_2O$, which represent key carbon and nitrogen fluxes. One example pattern of these three variables are shown in Figure 1. Due to the difficulties in field data collection, many other variables are often less available, though they are also included in our simulated data. Additionally, we also include driver variables for agricultural ecosystems, e.g., weather, soil, and management, in both simulated and observed data, thereby facilitating model development. In the following, we will provide details for these data sources. The statistics and distributions of every feature are provided in Appendix A.1.

### 3.1    SIMULATED DATA

We create simulated data using two PBMs, Ecosys (Grant, 2001) and Daycent (Del Grosso et al., 2001; 2005). Both PBMs are parameterized to reflect real agricultural processes, providing a valuable testing ground for model's capabilities for capturing underlying complex dynamics. Specifically, the Ecosys model has been well calibrated/validated using flux tower observations (Zhou et al., 2021), and has demonstrated its performance in simulating crop yield, soil biogeochemistry, GHG emissions, and nutrient losses in response to cover crop and fertilizer management (Qin et al., 2021; Li et al., 2022), tile drainage (Ma et al., 2021), and climate variabilities (Yang et al., 2022). DayCent® (version 279), which is used in this study, has been applied to simulate cropland and grassland systems at scales ranging from farm-level greenhouse-gas accounting (e.g., COMET-Farm™ [1]) and carbon crediting (Mathers et al., 2023), to serving as the Tier-3 process-based model for the national greenhouse-gas inventory (EPA, 2021). It has also been used to project soil-based carbon removal under various agricultural conservation practices for the U.S. "Roads to Removal" report (Pett-Ridge et al., 2023). Moreover, the strength of simulated data is in that they can mimic agricultural dynamics under a variety of scenarios, e.g., different fertilization rates, to reflect variability in human management practices.

---

[1]https://comet-farm.com/home

### 3.1.1 Simulation by Ecosys

Ecosys follows detailed biophysical and biogeochemical rules (Grant, 2001; Zhou et al., 2021), and encompasses a comprehensive set of input drivers (i.e., features) related to weather, soil, and management practices. Specifically, the input driver variables include:

- **Weather:** Daily maximum and minimum air temperature (TMAX, TMIN, units are °C), precipitation (PREC, mm day$^{-1}$), radiation (RADN), maximum and minimum humidity (HUMIDITY), and wind speed (WIND).
- **Soil:** Soil bulk density (TBKDS), sand content (TCSAND, g kg$^{-1}$), silt content (TCSILT, g kg$^{-1}$), soil pH (TPH, unitless), and soil organic carbon content (TSOC, g C kg$^{-1}$).
- **Management:** Fertilizer application rate (FERTZR_N, g N m$^{-2}$), planting day of the year (PDOY, day), and plant type (PLANTT, 1 for corn and 0 for soybean).

The output variables of Ecosys are categorized into four primary groups:

- **Carbon:** Ecosystem respiration (Reco, g C m$^{-2}$ day$^{-1}$), net ecosystem exchange (NEE, g C m$^{-2}$ day$^{-1}$), gross primary productivity (GPP, g C m$^{-2}$ day$^{-1}$), crop yield (Yield, kg ha$^{-1}$ year$^{-1}$), change in soil organic carbon ($\Delta$SOC, g C m$^{-2}$ year$^{-1}$), and leaf area index (LAI, fraction).
- **Nitrogen:** Nitrous oxide flux (N$_2$O, g N m$^{-2}$ day$^{-1}$), ammonium concentration ([NH$_4^+$], g Mg$^{-1}$) at different soil layers (5 cm, 20 cm, 30 cm), and nitrate concentration ([NO$_3^-$], g Mg$^{-1}$) at similar depths.
- **Water:** Soil water content (SWC, m$^3$ m$^{-3}$) at different layers (5 cm, 20 cm, 30 cm) and evapotranspiration (ET, mm day$^{-1}$).
- **Thermal:** Daily maximum and minimum soil temperature (Tsoil_max, Tsoil_min, °C) at different layers (5 cm, 20 cm, 30 cm).

For the Ecosys model, we generate daily synthetic data for the period 2000–2018 across 99 randomly selected counties in Iowa, Illinois, and Indiana states of USA. To reflect the variability in management practices, the simulation is performed using 20 different nitrogen (N) fertilization rates ranging from 0 to 33.6 g N m$^{-2}$ in each county. Such variation in fertilization is captured by the driver variable FERTZR_N, while all other driver variables are set as true values for the corresponding locations and dates. We represent the Ecosys data as $\mathcal{D}_{es} = \{\mathbf{X}_{es}, \mathbf{Y}_{es}\}$. The structure of the input and output variables is $\mathbf{X}_{es} \in \mathbb{R}^{N_{es} \times T_{es} \times D^{in} \times S_{es}}$ and $\mathbf{Y}_{es} \in \mathbb{R}^{N_{es} \times T_{es} \times D_{es}^{out} \times S_{es}}$, where $N_*$, $T_*$, $\{D^{in}, D_*^{out}\}$, and $S_*$ represent the number of sampled locations, the time span (at daily scale), the number of input variables and output variables, and different simulation scenarios, used in the simulation, respectively.

### 3.1.2 Simulation by DayCent

The biogeochemical model DayCent is currently used by the Environmental Protection Agency (EPA) and United States Department of Agriculture (USDA) for the U.S. national inventory of agricultural GHG emissions (Necpálová et al., 2015; Del Grosso et al., 2020). This model uses the same set of input features as Ecosys (i.e., $\mathbf{X}_{es}$), but produces a slightly different set of output variables. Specifically, the output variables of DayCent include:

- **Carbon:** Similar to Dataset Ecosys, it covers Reco, NEE, GPP, yield, $\Delta$SOC, and LAI.
- **Nitrogen:** This dataset measures N$_2$O and provides ammonium concentrations ([NH$_4^+$]) only from the top 10 cm of soil, lacking measurements at deeper layers. Nitrate concentrations ([NO$_3^-$]) remain consistent across layers.
- **Water:** SWC and ET data are available with no negative values for ET, indicating atmospheric fluxes.
- **Thermal:** Soil temperatures are measured consistently at specified depths.

For the DayCent model, simulations are conducted daily for 2,562 sites randomly sampled in the midwestern United States from 2000 to 2020. Each site is modeled under 42 scenarios, with varying nitrogen (N) fertilizer rates from 0 to 33.6 g N m$^{-2}$, different fertilization timing (e.g., at planting or 30 days after), and crop rotation (corn-soybean or soybean-corn). We represent the DayCent data as $\mathcal{D}_{dc} = \{\mathbf{X}_{dc}, \mathbf{Y}_{dc}\}$, where $\mathbf{X}_{dc} \in \mathbb{R}^{N_{dc} \times T_{dc} \times D^{in} \times S_{dc}}$ and $\mathbf{Y}_{dc} \in \mathbb{R}^{N_{dc} \times T_{dc} \times D_{dc}^{out} \times S_{dc}}$

## 3.2 OBSERVATIONAL DATA

The $N_2O$ observations are collected from a controlled-environment facility using soil samples from a corn-soybean rotation farm (Liu et al., 2022a). Six chambers were used to grow continuous corn during 2016-2018, with precise monitoring of $N_2O$ fluxes in response to different precipitation treatments from April 1st to July 31st. $N_2O$ fluxes were measured hourly and processed for a daily time scale, alongside soil moisture, nitrate, ammonium concentrations, and environmental variables. Simulating agricultural $N_2O$ emissions is important for mitigating climate change but challenging due to its hot moment/spot and underlying complex biogeochemical processes.

The observational data for $CO_2$ fluxes and GPP are collected from 11 cropland eddy covariance (EC) flux tower sites located in major U.S. corn and soybean production regions (Liu et al., 2024a). These sites, including US-Bo1, US-Bo2, US-Br1, US-Br3, US-IB1, US-KL1, US-Ne1, US-Ne2, US-Ne3, US-Ro1, and US-Ro5, span across Illinois, Iowa, Michigan, Nebraska, and Minnesota states. The GPP data was decomposed from observed $CO_2$ fluxes at these sites using the ONEFlux tool. The weather data were retrieved from the EC flux towers, while soil information and plant type information were retrieved from gSSURGO, and CDL data. The dataset provided daily time scale measurements, covering a time span from 2000 to 2020 across 11 cropland EC flux tower sites, with each site having different operational periods, ranging from 5 to 19 years. The dataset provides comprehensive temporal and spatial variance for validating ML models's ability to capture agricultural carbon flux dynamics.

To facilitate model construction, we also include driver variables from the locations and time periods for the observation samples. Hence, we can represent the observation data for each variable as $\mathcal{D}^v = \{\mathbf{X}^v, \mathbf{Y}^v\}$, where $\mathbf{X}^v \in \mathbb{R}^{N^v \times T^v \times D^{\text{in}}}$ and $\mathbf{Y}^v \in \mathbb{R}^{N^v \times T^v}$, and $v$ denotes $N_2O$, $CO_2$, or GPP.

## 3.3 ML-READY DATA FORMATS

All data underwent a series of preprocessing steps to ensure compatibility and quality. The input features are normalized to facilitate the model's learning process. Observational data have missing values as shown in Figure 1. We apply masks to exclude missing values from loss calculations during training and evaluate the performance only on available observations. For both simulated and observation data, we cut the original sequence of $T$ dates into yearly sub-sequences of length 365 for the ease of model learning.

# 4 AGROFLUX: EVALUATION FRAMEWORK

Effective models for monitoring agricultural ecosystems are expected to generalize well to unseen scenarios, including unseen time periods or different spatial locations. In real-world agricultural applications, these models are tasked with predicting output variables (e.g., GPP, $CO_2$, and $N_2O$) based on input drivers that can be readily observed. To standardize this evaluation, AgroFlux establishes benchmark scenarios and tasks that capture the main challenges of agricultural flux prediction. These benchmark settings provide a unified framework for testing both model generalization and transfer learning techniques across simulated and observational datasets.

## 4.1 BENCHMARK SCENARIOS

AgroFlux evaluates models in two generalization scenarios: temporal extrapolation and spatial extrapolation.

**Temporal extrapolation.** Models are trained on data from past years and tested on data from later years, mimicking realistic forecasting where historical records are used to predict future outcomes within the same sites. This setup prevents data leakage and ensures the evaluation reflects predictive capability across time.

**Spatial extrapolation.** Models are trained and tested on data from different locations, assessing the ability to generalize across heterogeneous soil, weather, and management conditions. This is important because observations are often limited to specific sites with flux towers or chambers. We adopt multi-fold cross-validation to provide a comprehensive assessment of spatial generalization.

### 4.2 BENCHMARK TASKS AND PROTOCOLS

Building on these two scenarios, AgroFlux defines three benchmark tasks with fixed train/validation/test splits. These standardized protocols ensure fair comparison across models and transfer learning techniques.

**Predicting simulated data.** For Ecosys temporal extrapolation, we use data from 2000–2012 across all locations as training, 2013–2015 as validation, and 2016–2018 as testing. For Ecosys spatial extrapolation, all 99 locations are divided into five equal folds, training on four folds and testing on the remaining fold. For DayCent, we use 2000–2016 as training, 2017–2018 as validation, and 2019–2020 as testing for temporal extrapolation, with a similar five-fold split for spatial extrapolation.

**Predicting observational data.** For $N_2O$ temporal extrapolation, we train on 2016–2017 and test on 2018. For $CO_2$ and GPP, we train on 2000–2015 and test on 2016–2020. This temporal split strategy ensures that no future information leaks into training. For spatial extrapolation, we conduct five-fold cross-validation across all available sites: given $N^v$ locations, each fold takes $\lfloor N^v/5 \rfloor$ for testing and the remainder for training.

**Transfer learning.** In this benchmark task, each model is first pretrained on simulated data (Ecosys or DayCent) and then fine-tuned on observational datasets. We adopt the same splits as in the observational data prediction tasks to ensure comparability, making this a controlled benchmark for evaluating transfer learning effectiveness.

### 4.3 EVALUATION METRICS

To systematically assess model performance in predicting agricultural carbon and nitrogen fluxes, AgroFlux specifies three complementary metrics as the benchmark metrics: coefficient of determination ($R^2$), root mean squared error (RMSE), and mean absolute error (MAE). $R^2$ captures explained variance, RMSE penalizes large deviations, and MAE reflects average prediction error.

## 5 BASELINE RESULTS

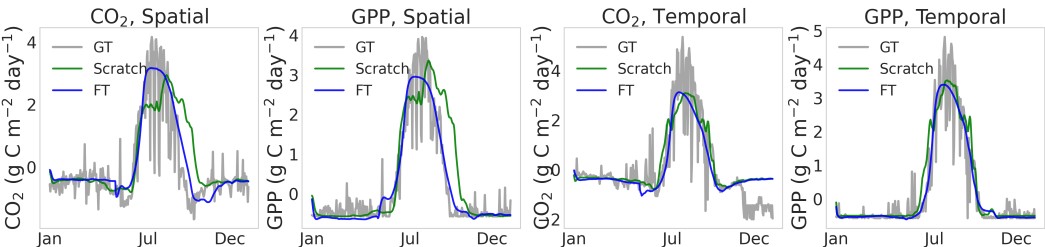

Figure 2: Comparison of LSTM predictions: trained from scratch versus fine-tuned (FT).

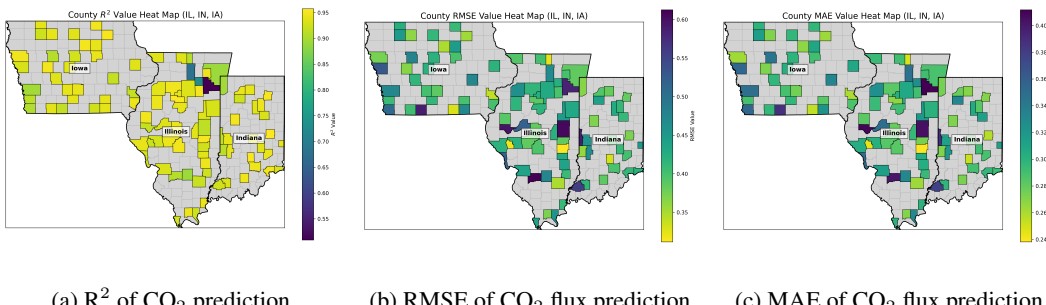

(a) $R^2$ of $CO_2$ prediction    (b) RMSE of $CO_2$ flux prediction    (c) MAE of $CO_2$ flux prediction

Figure 3: LSTM prediction performance on the Ecosys dataset averaged from 2016 to 2018.

## 5.1 EXPERIMENTAL SETTINGS

We establish benchmark baselines by training and evaluating six deep learning models for agricultural carbon and nitrogen flux prediction: LSTM (Hochreiter & Schmidhuber, 1997), EA-LSTM (Li et al., 2018), TCN (Lea et al., 2016), Transformer (Vaswani et al., 2017), iTransformer (Liu et al., 2024b), and Pyraformer (Liu et al., 2022b). Standard LSTM and EA-LSTM are commonly used in ecological modeling. We also test the TCN model, which leverages causal and dilated convolutions to model temporal dependencies. Additionally, we evaluate three Transformer-based models: standard Transformer, iTransformer, and Pyraformer. The iTransformer focuses on inter-variable dependencies by applying self-attention across variables rather than time steps, while Pyraformer employs a pyramid attention mechanism that captures multi-resolution temporal dependencies. Implementation details of these models can be found in Appendix A.3.

For the transfer learning baselines, we used two technqiues—pretrain-finetune and adversarial training on each model. Pretrain-finetune involves first training a model on either the Ecosys or DayCent simulated data and then fine-tuning the pre-trained model using the corresponding observational data ($CO_2$ flux, GPP, or $N_2O$ flux). In addition to pretrain–finetune, we also evaluates adversarial training technique. In this approach, a domain discriminator is trained jointly with the prediction model to distinguish whether a feature representation is from simulated or observational data. The prediction model is trained not only to minimize prediction loss but also to maximize the error from the discriminator, which allows domain-invariant feature learning.

Table 1: Model evaluation results on simulated datasets in $R^2$

| Model | DayCent | | | | | | Ecosys | | | | | |
| | Spatial | | | Temporal | | | Spatial | | | Temporal | | |
| | $CO_2$ | GPP | $N_2O$ | $CO_2$ | GPP | $N_2O$ | $CO_2$ | GPP | $N_2O$ | $CO_2$ | GPP | $N_2O$ |
|---|---|---|---|---|---|---|---|---|---|---|---|---|
| EALSTM | 0.819 | 0.846 | 0.096 | 0.810 | 0.838 | 0.180 | 0.882 | 0.887 | 0.253 | 0.872 | 0.835 | 0.275 |
| iTransformer | 0.846 | 0.851 | 0.112 | 0.823 | 0.852 | 0.152 | 0.857 | 0.908 | 0.516 | 0.740 | 0.603 | -1.009 |
| LSTM | 0.852 | 0.865 | 0.139 | 0.846 | **0.859** | **0.220** | 0.937 | 0.944 | 0.676 | **0.938** | **0.890** | **0.697** |
| Pyraformer | **0.896** | **0.903** | 0.211 | 0.836 | 0.841 | 0.123 | 0.861 | 0.937 | 0.547 | 0.757 | 0.804 | 0.190 |
| TCN | 0.744 | 0.776 | 0.100 | 0.765 | 0.782 | 0.192 | 0.895 | 0.795 | 0.392 | 0.868 | 0.712 | 0.476 |
| Transformer | 0.891 | 0.898 | **0.226** | **0.850** | 0.857 | 0.117 | **0.953** | **0.971** | **0.732** | 0.929 | 0.878 | 0.551 |

Table 2: Model evaluation results on observation datasets in $R^2$

| Model | $CO_2$ | | GPP | | $N_2O$ | |
| | Spatial | Temporal | Spatial | Temporal | Spatial | Temporal |
|---|---|---|---|---|---|---|
| EALSTM | 0.041 | 0.038 | 0.089 | 0.048 | -0.005 | -0.082 |
| iTransformer | **0.563** | 0.707 | **0.656** | 0.808 | 0.471 | 0.171 |
| LSTM | 0.339 | 0.552 | 0.503 | 0.730 | -0.009 | -0.148 |
| Pyraformer | 0.539 | 0.745 | 0.647 | 0.858 | **0.883** | 0.243 |
| TCN | 0.341 | 0.551 | 0.498 | 0.725 | 0.394 | 0.223 |
| Transformer | 0.535 | **0.784** | 0.642 | **0.869** | 0.560 | **0.433** |

## 5.2 PREDICTING SIMULATED DATA

We report benchmark baselines for six deep learning models on the DayCent and Ecosys datasets. Tables 1, 5, and 6 present the $R^2$, RMSE, and MAE results for both spatial and temporal extrap-

Table 3: Transfer learning results by $R^2$ (pretrain-finetune)

| Model | Source: DayCent | | | | | | Source: Ecosys | | | | | |
| | Spatial | | | Temporal | | | Spatial | | | Temporal | | |
| | $CO_2$ | GPP | $N_2O$ | $CO_2$ | GPP | $N_2O$ | $CO_2$ | GPP | $N_2O$ | $CO_2$ | GPP | $N_2O$ |
|---|---|---|---|---|---|---|---|---|---|---|---|---|
| EALSTM | 0.351 | 0.505 | 0.065 | 0.689 | 0.777 | 0.208 | -0.439 | **0.644** | -0.028 | 0.301 | 0.772 | -0.030 |
| iTransformer | 0.486 | 0.622 | -0.324 | 0.617 | 0.770 | -0.236 | 0.128 | 0.447 | -0.324 | -0.001 | 0.759 | -0.236 |
| LSTM | **0.648** | **0.750** | 0.515 | 0.727 | **0.855** | 0.399 | 0.089 | 0.639 | 0.736 | 0.727 | **0.854** | 0.207 |
| Pyraformer | 0.536 | 0.668 | 0.521 | 0.704 | 0.822 | **0.573** | 0.038 | 0.578 | **0.754** | 0.731 | 0.844 | **0.661** |
| TCN | 0.538 | 0.689 | **0.673** | **0.731** | 0.835 | 0.434 | **0.408** | 0.643 | 0.666 | **0.752** | 0.840 | 0.357 |
| Transformer | 0.424 | 0.602 | 0.007 | 0.620 | 0.732 | -0.241 | 0.406 | 0.595 | 0.013 | 0.615 | 0.742 | -0.230 |

Table 4: Transfer learning results by $R^2$ (adversarial training)

| Model | Source: DayCent | | | | | | Source: Ecosys | | | | | |
| | Spatial | | | Temporal | | | Spatial | | | Temporal | | |
| | $CO_2$ | GPP | $N_2O$ | $CO_2$ | GPP | $N_2O$ | $CO_2$ | GPP | $N_2O$ | $CO_2$ | GPP | $N_2O$ |
|---|---|---|---|---|---|---|---|---|---|---|---|---|
| EALSTM | 0.551 | 0.707 | 0.195 | 0.700 | 0.814 | 0.332 | 0.404 | 0.681 | -0.023 | 0.672 | 0.823 | **0.349** |
| iTransformer | 0.472 | 0.637 | -0.324 | 0.622 | 0.768 | -0.236 | 0.476 | 0.601 | -0.324 | -0.001 | 0.757 | -0.236 |
| LSTM | **0.666** | **0.781** | 0.625 | **0.731** | **0.840** | 0.229 | **0.647** | **0.790** | 0.803 | **0.757** | **0.869** | 0.076 |
| Pyraformer | 0.561 | 0.662 | **0.824** | 0.702 | 0.808 | **0.367** | 0.364 | 0.527 | **0.875** | 0.732 | 0.834 | 0.271 |
| TCN | 0.486 | 0.638 | 0.668 | 0.633 | 0.827 | **0.367** | 0.422 | 0.612 | 0.649 | 0.717 | 0.809 | 0.317 |
| Transformer | 0.422 | 0.594 | 0.023 | 0.627 | 0.734 | -0.184 | 0.385 | 0.524 | 0.059 | 0.649 | 0.757 | -0.192 |

olation tasks across three key variables: $CO_2$ flux, GPP, and $N_2O$ flux. Several key observations emerge from the results.

*Model performance varies across simulated datasets:* All models generally perform better on the Ecosys dataset compared to the DayCent dataset, particularly for $N_2O$ flux prediction. For instance, LSTM achieves an $R^2$ of 0.697 for $N_2O$ flux spatial extrapolation on Ecosys, while only reaching 0.220 on DayCent. These baselines show that even with strong models, DayCent remains more challenging while Ecosys simulations contain more consistent and learnable patterns, highlighting the benchmark's difficulty, especially for certain processes, e.g., nitrogen cycles.

*LSTM performs strongly on temporal extrapolation:* LSTM achieves the best results for all three variables in the Ecosys temporal setting, reaching $R^2$ values of 0.938 for $CO_2$ flux, 0.890 for GPP, and 0.697 for $N_2O$ flux. In addition, LSTM These results achieves the best results for two out of three variables in the Daycent temporal setting as well. These results highlight the ability of recurrent structures to capture long-term temporal dynamics of agroecosystem fluxes under varying environmental drivers. While performance varies in other settings, the Ecosys temporal baselines from LSTM provide useful reference points for future temporal modeling approaches.

*Transformer family performs strongly on spatial extrapolation:* Within the DayCent setting, Pyraformer achieves the best $R^2$ for $CO_2$ flux(0.896) and GPP (0.903), while the standard Transformer provides the strongest performance for $N_2O$ flux(0.226). In the Ecosys setting, Transformer dominates across all spatial variables, reaching 0.953 for $CO_2$ flux, 0.971 for GPP, and 0.732 for $N_2O$ flux. These results highlight that attention-based architectures are particularly effective for spatial generalization of carbon and nitrogen fluxes, with different Transformer variants excelling under different conditions.

*$N_2O$ flux prediction remains challenging:* Across all models and datasets, $N_2O$ flux prediction consistently shows lower $R^2$ scores compared to $CO_2$ flux and GPP prediction. This reflects the inherent complexity and higher variability of $N_2O$ flux dynamics, which are known to be influenced by management practices that are not fully captured by input driver variables. More specifically, $N_2O$ flux has hot moments in time and hot spots in space due to being driven by a series of biotic and abiotic processes related to the soil microbial, N concentration and water-thermal conditions, and various management practices, such as fertilization, irrigation, tillage, and historical land uses, which are often not available. Therefore, the highly nonlinear nature of complex processes and incomplete information of management make the prediction of N dynamics challenging.

It is worth mentioning that we also established baselines for a different training schema, where we train specialized models for each output variable group. Results can be found in Appendix A.4.

## 5.3 PREDICTING OBSERVATIONAL DATA

We report baseline benchmark results for $N_2O$ flux, $CO_2$ flux, and GPP observational datasets in Tables 2, 7, and 8. For $N_2O$ prediction, Pyraformer demonstrates exceptional performance, achieving the highest $R^2$ of 0.883 for spatial extrapolation, while Transformer achieves the best temporal performance with an $R^2$ of 0.433. Other models struggle considerably, with several producing negative $R^2$ values, highlighting the inherent challenges in modeling these episodic emissions. These baselines demonstrate that AgroFlux remains challenging even for state-of-the-art temporal models.

For carbon features ($CO_2$ flux and GPP), Transformer achieves the strong results across both spatial and temporal extrapolation tasks. It outperforms other models with $R^2$ values of 0.784 for $CO_2$ flux

temporal prediction and 0.869 for GPP temporal prediction. iTransformer also shows competitive performance in spatial extrapolation, achieving the best $R^2$ values of 0.563 for $CO_2$ flux and 0.656 for GPP. While Pyraformer remains competitive, the results indicate that different Transformer variants excel under different extrapolation scenarios.

## 5.4 Transfer Learning from Simulations to Observations

We also provide benchmark baselines for transfer learning. We explore two strategies: pretrain-finetune and adversarial learning. Table 3 reports finetuning results, while Table 4 presents adversarial learning results.

We compare the results of transfer learning techniques to the baseline results from Section 5.3. The pretrain-finetune approach consistently improves model performance for $CO_2$ flux and GPP. For example, when using DayCent as source, LSTM improves from 0.339 to 0.648 for $CO_2$ spatial prediction, and from 0.503 to 0.855 for GPP spatial prediction. Similar gains are observed in temporal extrapolation, where LSTM increases from 0.552 to 0.727 for $CO_2$ and from 0.730 to 0.855 for GPP. EA-LSTM also benefits, reaching 0.689 and 0.777 for $CO_2$ and GPP temporal prediction, respectively. With Ecosys as source, Pyraformer achieves strong results for $N_2O$, with $R^2$ of 0.754 (spatial) and 0.661 (temporal). However, improvements for $N_2O$ are generally limited compared to $CO_2$ and GPP, reflecting the much smaller size of the observational $N_2O$ dataset.

The adversarial training strategy further boosts performance in several cases. For example LSTM rises from 0.563 to 0.666 for $CO_2$ spatial and from 0.503 to 0.781 for GPP spatial using DayCent as source. These results show that adversarial alignment can more effectively bridge the distribution gap between simulations and observations.

Overall, these experiments establish the first transfer learning baselines for agricultural flux prediction, highlighting both the promise of simulation-to-observation transfer and the persistent challenge of $N_2O$ flux. Future models can use these baselines for systematic comparison.

## 6 Conclusion

This paper introduces AgroFlux, the first benchmark suite for agricultural greenhouse gas (GHG) flux prediction. AgroFlux defines standardized benchmark scenarios and tasks (simulated prediction, observational prediction, and transfer learning), along with consistent evaluation metrics ($R^2$, RMSE, MAE). These elements together provide a unified framework for fair comparison of machine learning methods.We also establish baseline results from six ML models and two transfer learning techniques, which serve as reference points for future uses on this benchmark. Our baseline results shows taht AgroFlux remains a challenging benchmark, especially for $N_2O$ flux prediction. By providing both comprehensive data and standardized evaluation protocols, AgroFlux transforms agricultural flux prediction into a benchmark challenge. We expect it to catalyze the development of new algorithms and analytical tools, enable reproducible comparison through leaderboards, and ultimately support more informed and timely decision-making in agricultural management and climate mitigation strategies.

**Limitations and Future Work** While AgroFlux provides a first-of-its-kind benchmark for agricultural GHG flux prediction, several limitations remain. First, the observational datasets, especially for $N_2O$ flux, have limited temporal duration and spatial coverage compared to the simulated data. This scarcity of high-quality measurements constrains the ability of models to fully capture and evaluate long-term generalization. We plan to continually expand AgroFlux by incorporating newly collected flux measurements and agricultural variables across more diverse regions, soil types, and management practices. This will facilitate both robust model training and more comprehensive evaluation protocols.

In addition, our benchmark currently evaluates deep learning models under fixed experimental splits and a limited set of metrics ($R^2$, RMSE, MAE). Future extensions could explore broader evaluation protocols, such as uncertainty quantification, robustness to missing or noisy inputs, and long-term stability under distributional shifts. Incorporating these aspects will make AgroFlux an even stronger platform for developing generalizable, reliable, and interpretable models for agroecosystem GHG prediction.

ETHICS STATEMENT

We acknowledge and commit to the ICLR Code of Ethics. All data licenses and permissions are respected, and we release the dataset and code under an open license.

REPRODUCIBILITY STATEMENT

All dataset construction, preprocessing steps, and fixed train/validation/test splits are documented in the main text and Appendix. Implementation details, model architectures, hyperparameters, and training protocols are provided in Appendix A.3. We include the complete code in the supplementary material, and also provide anonymized public links to the source code: avaliable upon acceptance. Due to size limit of the supplementary material and the anonymity requirement, the link to the dataset card on HuggingFace will be made available upon acceptance.

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

# A APPENDIX

## A.1 DATA DISTRIBUTIONS

All data distribution and statistic can be found in Figure 4, 5, and 6.

## A.2 EVALUATION METRICS

To systematically assess model performance in predicting agricultural carbon and nitrogen fluxes, we employ three complementary evaluation metrics:

**Coefficient of determination** ($R^2$) measures the proportion of variance in the target variable that is predictable from the model. The value of $R^2$ ranges from 0 to 1, with higher values indicating better performance. $R^2$ is particularly useful for understanding how well the model captures the temporal and spatial patterns in the flux data.

**Root mean squared error (RMSE)** quantifies the standard deviation of prediction vs. true values. RMSE is sensitive to large errors and provides a measure of prediction accuracy in the same units as the target variable (e.g., g C $m^{-2}day^{-1}$ for carbon fluxes). Lower RMSE values indicate better model performance.

**Mean absolute error (MAE)** represents the average of the absolute differences between predictions and actual observations. MAE is less sensitive to outliers compared to RMSE and provides a straightforward interpretation of average model error. Like RMSE, lower values indicate better performance.

## A.3 IMPLEMENTATION DETAILS

For all models, we use a hidden dimension of 50, a learning rate of $1e-3$ with the Adam optimizer, and a dropout rate of 0.2. The LSTM and EA-LSTM models used 3 layers, while Transformer-based models employed 3 encoder layers, 1 decoder layer, and 5 attention heads. For TCN, we used 3 temporal blocks with a kernel size of 5. Batch sizes are set to 256 for standard models and 10 for the iTransformer. All methods are from Time Series Library (TSLib) (Wu et al., 2023; Wang et al., 2024) and run on a RTX 5080 GPU.

## A.4 PREDICTING SIMULATED DATA USING SPECIALIZED MODELS

Figure 7 illustrates the comparison between two distinct training approaches: training separate models for each output category versus training a single comprehensive model. As mentioned in section 3.1, the output variables in the simulated data can be categorized into four groups - Carbon, Nitrogen, Water, and Thermal. This experiment evaluates whether training four specialized models (one for each output category) produces different results than training a single unified model with

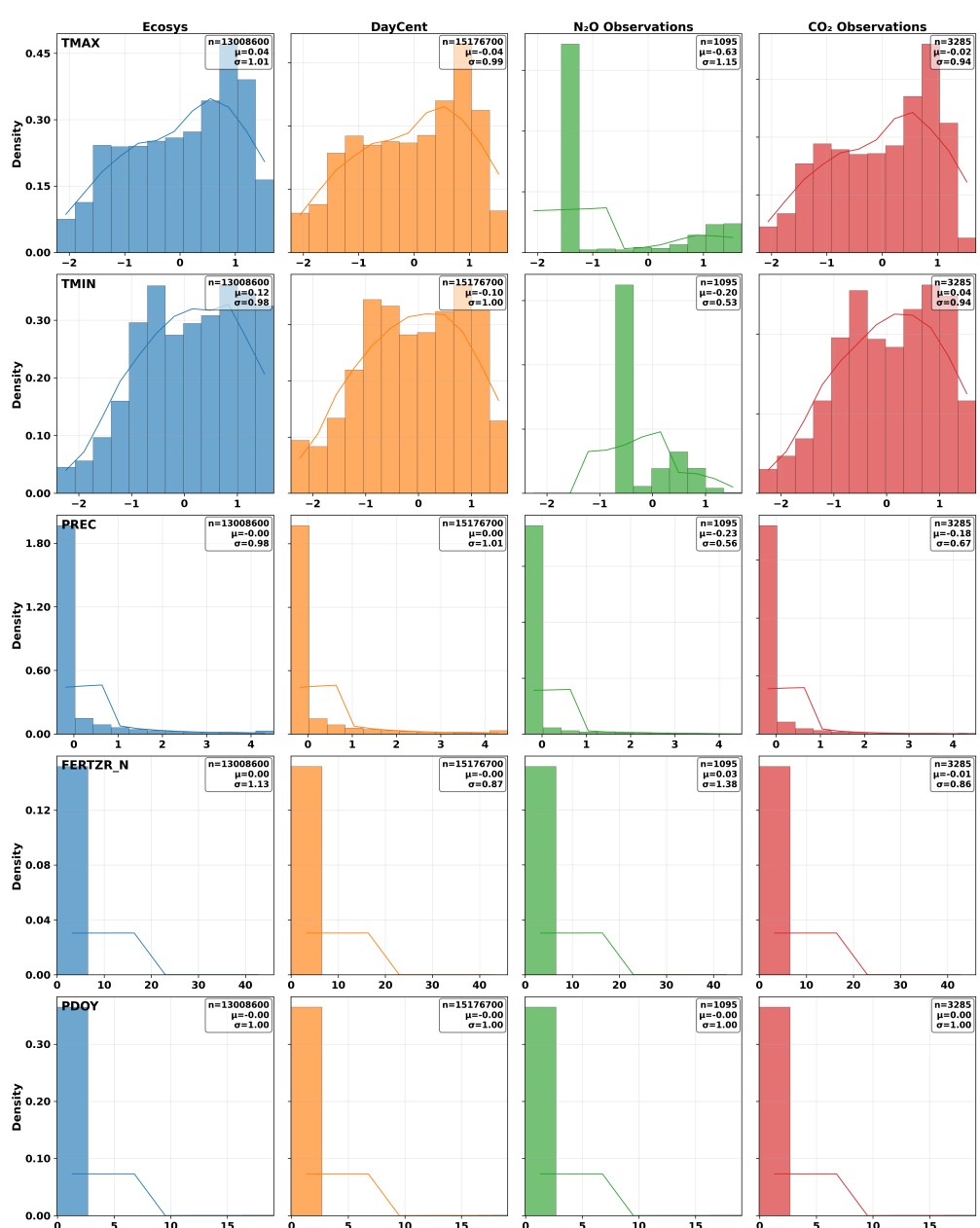

Figure 4: Feature distributions for the first five input features (TMAX, TMIN, PREC, FERTZR_N, PDOY) across four datasets (Ecosys, DayCent, $N_2O$ Observations, $CO_2$/GPP Observations as the columns). For each subplot, the X-axis represents value intervals, and Y-axis represents frequencies. Values are standardized and clipped to the 1st–99th percentile, and legends report n, mean, and std.

all output variables. The findings demonstrate that training separate models for each output feature group yields similar performance to training one comprehensive model that handles all output features simultaneously.

## A.5 EXPERIMENTS RESULTS

In this section, we present all evaluation result in section

Table 5: Model evaluation results on simulated datasets in RMSE

| Model | DayCent | | | | | | Ecosys | | | | | |
| | Spatial | | | Temporal | | | Spatial | | | Temporal | | |
| | $CO_2$ | GPP | $N_2O$ | $CO_2$ | GPP | $N_2O$ | $CO_2$ | GPP | $N_2O$ | $CO_2$ | GPP | $N_2O$ |
|---|---|---|---|---|---|---|---|---|---|---|---|---|
| EALSTM | 1.249 | 1.402 | **0.002** | 1.386 | 1.550 | 0.002 | 0.571 | 2.095 | 0.002 | 0.564 | 2.950 | 0.002 |
| iTransformer | 1.154 | 1.380 | **0.002** | 1.337 | 1.483 | 0.002 | 0.627 | 1.890 | **0.001** | 0.803 | 4.582 | 0.003 |
| LSTM | 1.132 | 1.312 | **0.002** | 1.249 | **1.449** | **0.001** | 0.417 | 1.478 | **0.001** | **0.392** | **2.413** | **0.001** |
| Pyraformer | **0.949** | **1.113** | **0.002** | 1.287 | 1.536 | 0.002 | 0.620 | 1.568 | **0.001** | 0.776 | 3.211 | 0.002 |
| TCN | 1.487 | 1.691 | **0.002** | 1.543 | 1.800 | 0.002 | 0.539 | 2.817 | 0.002 | 0.572 | 3.901 | 0.002 |
| Transformer | 0.970 | 1.140 | **0.002** | **1.232** | 1.455 | 0.002 | **0.359** | **1.057** | **0.001** | 0.418 | 2.538 | **0.001** |

Table 6: Model evaluation results on simulated datasets in MAE

| Model | DayCent | | | | | | Ecosys | | | | | |
| | Spatial | | | Temporal | | | Spatial | | | Temporal | | |
| | $CO_2$ | GPP | $N_2O$ | $CO_2$ | GPP | $N_2O$ | $CO_2$ | GPP | $N_2O$ | $CO_2$ | GPP | $N_2O$ |
|---|---|---|---|---|---|---|---|---|---|---|---|---|
| EALSTM | 0.772 | 0.787 | 0.001 | 0.859 | 0.876 | **0.000** | 0.400 | 1.213 | 0.001 | 0.406 | 1.641 | 0.001 |
| iTransformer | 0.782 | 0.930 | **0.000** | 0.891 | 0.919 | 0.001 | 0.461 | 1.312 | 0.001 | 0.615 | 3.191 | 0.002 |
| LSTM | 0.673 | 0.701 | **0.000** | **0.744** | **0.762** | **0.000** | 0.295 | 0.854 | **0.000** | **0.283** | 1.361 | **0.000** |
| Pyraformer | **0.574** | 0.650 | **0.000** | 0.776 | 0.834 | 0.001 | 0.476 | 1.038 | **0.000** | 0.569 | 1.867 | 0.001 |
| TCN | 0.925 | 0.994 | 0.001 | 0.989 | 1.028 | **0.000** | 0.370 | 1.851 | 0.001 | 0.412 | 2.530 | 0.001 |
| Transformer | 0.580 | **0.620** | **0.000** | 0.748 | 0.793 | **0.000** | **0.250** | **0.571** | **0.000** | 0.298 | **1.340** | **0.000** |

## A.6   LLM USAGE

LLMs were used only as a general-purpose writing assist tool. They did not play a significant role in this research.

Table 7: Model evaluation results on observation datasets in RMSE

| Model | $CO_2$ | | GPP | | $N_2O$ | |
|---|---|---|---|---|---|---|
| | Spatial | Temporal | Spatial | Temporal | Spatial | Temporal |
| EALSTM | 3.951 | 3.657 | 6.378 | 6.695 | 0.007 | 0.006 |
| iTransformer | **2.668** | 2.017 | **3.922** | 3.003 | 0.005 | 0.005 |
| LSTM | 3.281 | 2.494 | 4.711 | 3.569 | 0.007 | 0.006 |
| Pyraformer | 2.739 | 1.882 | 3.970 | 2.583 | **0.002** | 0.005 |
| TCN | 3.275 | 2.499 | 4.736 | 3.600 | 0.005 | 0.005 |
| Transformer | 2.751 | **1.732** | 3.998 | **2.480** | 0.005 | **0.004** |

Table 8: Model evaluation results on observation datasets in MAE

| Model | $CO_2$ | | GPP | | $N_2O$ | |
|---|---|---|---|---|---|---|
| | Spatial | Temporal | Spatial | Temporal | Spatial | Temporal |
| EALSTM | 2.734 | 2.424 | 4.609 | 4.866 | 0.006 | 0.005 |
| iTransformer | 1.531 | 1.195 | 2.316 | 1.876 | 0.002 | **0.002** |
| LSTM | 1.984 | 1.401 | 2.897 | 1.883 | 0.006 | 0.005 |
| Pyraformer | 1.515 | 1.073 | **2.260** | 1.379 | **0.001** | 0.003 |
| TCN | 2.027 | 1.574 | 3.078 | 2.240 | 0.004 | 0.004 |
| Transformer | **1.494** | **0.976** | 2.435 | **1.284** | 0.002 | **0.002** |

Table 9: Transfer learning results for spatial experiments in RMSE (Adversarial Training)

| Model | DC | | | Ecosys | | |
|---|---|---|---|---|---|---|
| | $CO_2$ | GPP | $N_2O$ | $CO_2$ | GPP | $N_2O$ |
| EALSTM | 2.926 | 4.492 | 0.007 | 4.883 | 3.608 | 0.007 |
| iTransformer | 2.845 | 3.539 | 0.005 | 3.565 | 5.022 | 0.006 |
| LSTM | **2.355** | **3.476** | 0.005 | 3.467 | **3.505** | 0.004 |
| Pyraformer | 2.780 | 4.021 | **0.004** | 3.040 | 4.217 | **0.003** |
| TCN | 3.189 | 4.415 | 0.005 | 4.593 | 4.892 | 0.006 |
| Transformer | 2.830 | 4.152 | **0.004** | **2.759** | 3.929 | **0.003** |

Table 10: Transfer learning results for spatial experiments in MAE (Adversarial Training)

| Model | DC | | | Ecosys | | |
|---|---|---|---|---|---|---|
| | $CO_2$ | GPP | $N_2O$ | $CO_2$ | GPP | $N_2O$ |
| EALSTM | 1.738 | 2.559 | 0.004 | 2.832 | **2.145** | 0.005 |
| iTransformer | 1.666 | **2.101** | 0.004 | 1.956 | 3.629 | 0.005 |
| LSTM | **1.417** | 2.185 | 0.003 | 2.043 | 2.357 | **0.002** |
| Pyraformer | 1.574 | 2.297 | **0.002** | 1.769 | 2.460 | **0.002** |
| TCN | 1.998 | 2.825 | 0.004 | 2.737 | 2.926 | 0.004 |
| Transformer | 1.512 | 2.240 | **0.002** | **1.514** | 2.293 | **0.002** |

Table 11: Transfer learning results for temporal experiments in RMSE (Adversarial Training)

| Model | DC | | | Ecosys | | |
|---|---|---|---|---|---|---|
| | $CO_2$ | GPP | $N_2O$ | $CO_2$ | GPP | $N_2O$ |
| EALSTM | 1.996 | 3.158 | 0.005 | 2.758 | 3.292 | 0.006 |
| iTransformer | 1.920 | 2.675 | 0.005 | 1.988 | 3.369 | 0.006 |
| LSTM | 1.840 | **2.506** | 0.005 | 1.957 | 2.647 | 0.004 |
| Pyraformer | 1.972 | 2.781 | **0.004** | 1.927 | 2.685 | **0.003** |
| TCN | 2.490 | 3.544 | 0.005 | 2.624 | 3.617 | 0.005 |
| Transformer | **1.785** | 2.608 | **0.004** | **1.751** | **2.376** | 0.006 |

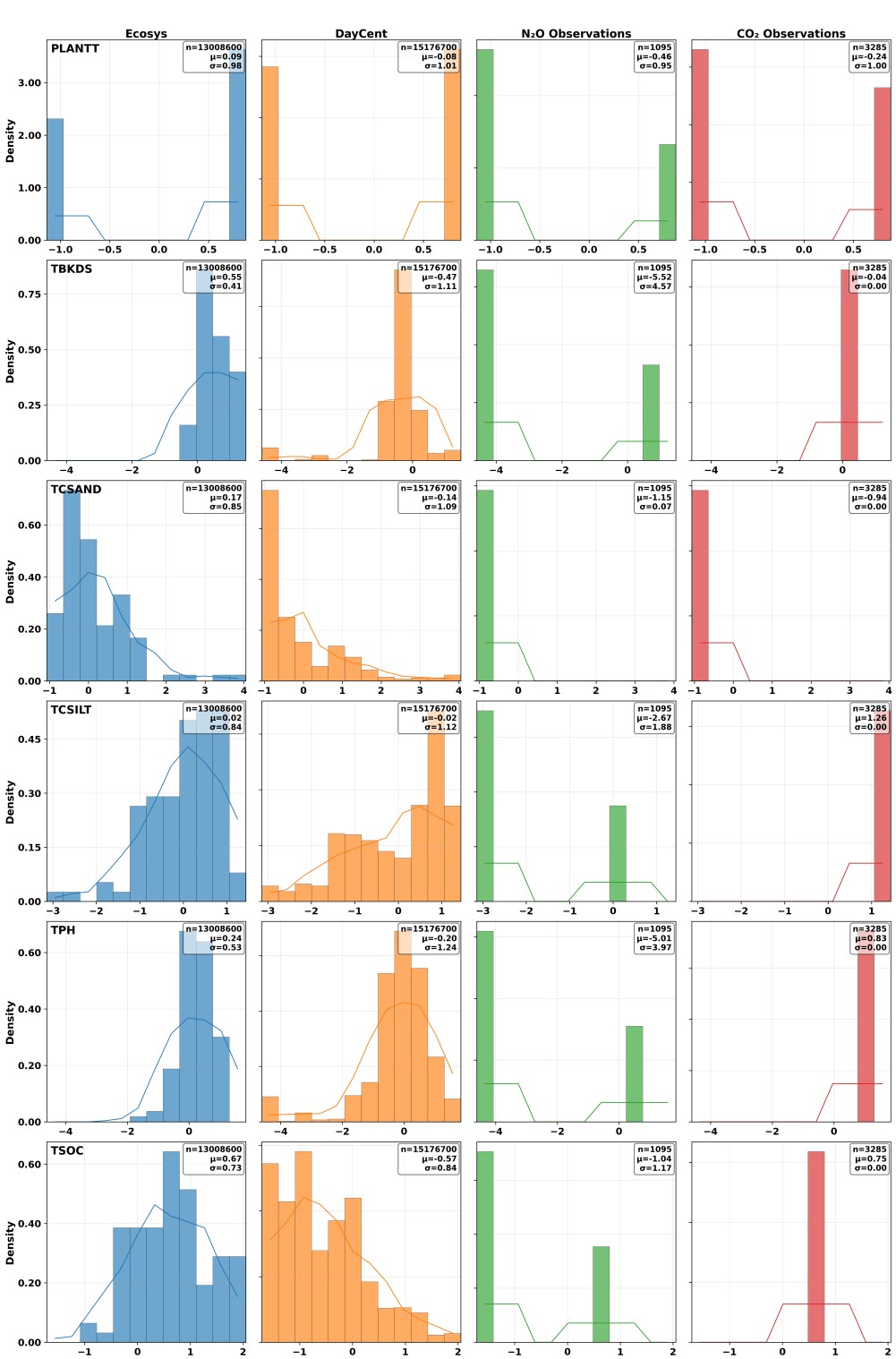

Figure 5: Feature distributions for the last six input features (PLANTT, TBKDS, TCSAND, TCSILT, TPH, TSOC) across four datasets (Ecosys, DayCent, $N_2O$ Observations, $CO_2$/GPP Observations as the columns). For each subplot, the X-axis represent value intervals, and Y-axis represent frequencies. Values are standardized and clipped to the 1st-–99th percentile, and legends report n, mean, and std.

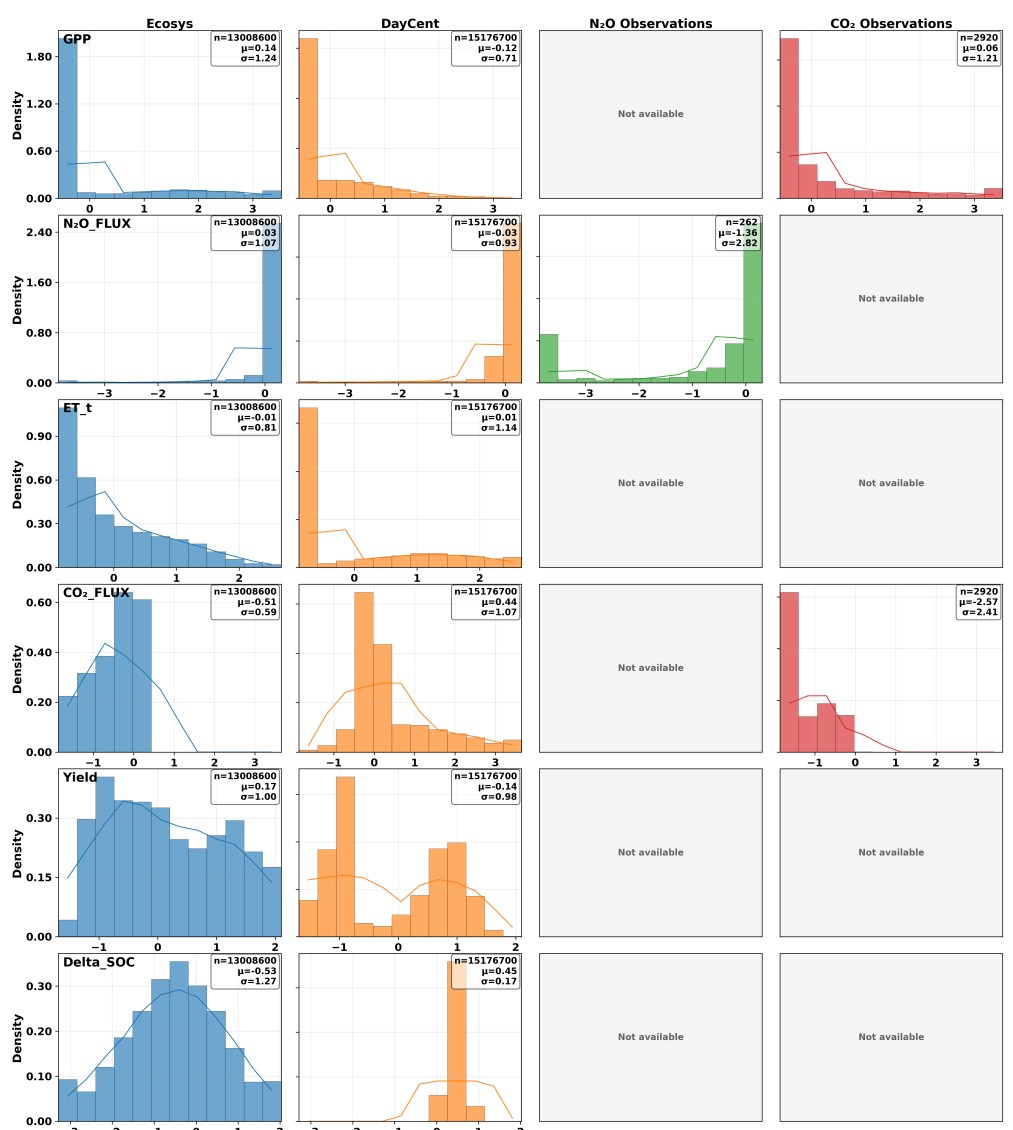

Figure 6: Feature distributions for the output features across four datasets (Ecosys, DayCent, $N_2O$ Observations, $CO_2$/GPP Observations as the columns). For each subplot, the X-axis represent value intervals, and Y-axis represent frequencies. Values are standardized and clipped to the 1st—99th percentile, and legends report n, mean, and std.

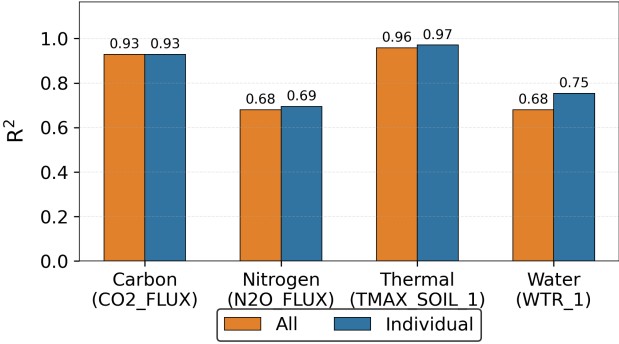

Figure 7: LSTM performance comparison between different training method for Ecosys.

Table 12: Transfer learning results for temporal experiments in MAE (Adversarial Training)

| Model | DC | | | Ecosys | | |
|---|---|---|---|---|---|---|
| | $CO_2$ | GPP | $N_2O$ | $CO_2$ | GPP | $N_2O$ |
| EALSTM | 1.148 | 1.705 | 0.003 | 1.654 | 1.807 | 0.004 |
| iTransformer | 1.137 | 1.504 | **0.002** | 1.158 | 1.867 | 0.005 |
| LSTM | 1.049 | **1.354** | **0.002** | 1.097 | 1.426 | **0.002** |
| Pyraformer | 1.144 | 1.482 | **0.002** | 1.106 | 1.431 | **0.002** |
| TCN | 1.550 | 2.093 | 0.004 | 1.600 | 2.104 | 0.003 |
| Transformer | **1.029** | 1.404 | **0.002** | **0.974** | **1.254** | 0.003 |

Table 13: Transfer learning results for spatial experiments in RMSE (Pretrain-finetune)

| Model | DC | | | Ecosys | | |
|---|---|---|---|---|---|---|
| | $CO_2$ | GPP | $N_2O$ | $CO_2$ | GPP | $N_2O$ |
| EALSTM | 2.535 | **3.374** | 0.006 | 3.403 | 4.231 | 0.007 |
| iTransformer | 2.709 | 3.756 | 0.004 | 2.898 | 4.690 | 0.005 |
| LSTM | 2.518 | 3.504 | 0.004 | **2.601** | **3.434** | 0.003 |
| Pyraformer | **2.476** | 3.769 | **0.003** | 3.039 | 4.276 | **0.002** |
| TCN | 3.165 | 4.520 | 0.004 | 3.250 | 4.657 | 0.005 |
| Transformer | 2.720 | 4.472 | 0.004 | 2.963 | 4.185 | 0.003 |

Table 14: Transfer learning results for spatial experiments in MAE (Pretrain-finetune)

| Model | DC | | | Ecosys | | |
|---|---|---|---|---|---|---|
| | $CO_2$ | GPP | $N_2O$ | $CO_2$ | GPP | $N_2O$ |
| EALSTM | **1.427** | **2.027** | 0.004 | 2.142 | 2.376 | 0.005 |
| iTransformer | 1.543 | 2.213 | 0.003 | 1.688 | 2.611 | 0.004 |
| LSTM | 1.453 | 2.034 | 0.002 | **1.521** | **2.066** | 0.002 |
| Pyraformer | 1.435 | 2.241 | **0.001** | 1.720 | 2.379 | **0.001** |
| TCN | 1.974 | 2.974 | 0.002 | 2.075 | 3.076 | 0.003 |
| Transformer | 1.557 | 2.501 | 0.002 | 1.681 | 2.557 | 0.002 |

Table 15: Transfer learning results for temporal experiments in RMSE (Pretrain-finetune)

| Model | DC | | | Ecosys | | |
|---|---|---|---|---|---|---|
| | $CO_2$ | GPP | $N_2O$ | $CO_2$ | GPP | $N_2O$ |
| EALSTM | 2.108 | 3.081 | 0.005 | 2.287 | 3.267 | 0.006 |
| iTransformer | 1.842 | 2.704 | **0.004** | 1.949 | 2.905 | 0.005 |
| LSTM | 1.818 | **2.439** | 0.006 | 1.939 | 2.656 | **0.004** |
| Pyraformer | 2.036 | 2.947 | **0.004** | 1.964 | 2.807 | **0.004** |
| TCN | 2.435 | 3.439 | 0.005 | 2.451 | 3.446 | 0.005 |
| Transformer | **1.701** | 2.442 | **0.004** | **1.680** | **2.439** | 0.005 |

Table 16: Transfer learning results for temporal experiments in MAE (Pretrain-finetune)

| Model | DC | | | Ecosys | | |
|---|---|---|---|---|---|---|
| | $CO_2$ | GPP | $N_2O$ | $CO_2$ | GPP | $N_2O$ |
| EALSTM | 1.186 | 1.642 | 0.003 | 1.454 | 1.783 | 0.005 |
| iTransformer | 1.078 | 1.540 | **0.002** | 1.101 | 1.650 | 0.003 |
| LSTM | 1.029 | 1.312 | 0.003 | 1.156 | 1.520 | **0.002** |
| Pyraformer | 1.194 | 1.645 | **0.002** | 1.137 | 1.548 | **0.002** |
| TCN | 1.495 | 2.017 | 0.003 | 1.493 | 2.022 | 0.003 |
| Transformer | **0.979** | **1.290** | **0.002** | **0.983** | **1.472** | 0.003 |

