# OpenReview forum: "AgroFlux: A Spatial-Temporal Benchmark for Carbon and Nitrogen Flux Prediction in Agricultural Ecosystems"
_ICLR.cc/2026/Conference — ICLR 2026 Conference Withdrawn Submission_

### Official Review · Reviewer_5EHZ · 2025-10-31

**Soundness:** 2
**Presentation:** 2
**Contribution:** 2
**Rating:** 4
**Confidence:** 4

**Summary:**

This paper presents AgroFlux, a proposed benchmark dataset and evaluation framework for agricultural carbon and nitrogen flux prediction. AgroFlux integrates simulated data generated from two process-based models (Ecosys and DayCent) with limited observational data from flux towers and controlled-environment experiments. The benchmark defines standardized tasks (temporal extrapolation, spatial extrapolation, and transfer learning), metrics (R², RMSE, MAE), and protocols for evaluating deep learning models, including LSTM, TCN, Transformer, Pyraformer, and iTransformer. The authors report extensive baseline results and suggest that AgroFlux can serve as a unifying testbed for AI-driven modeling of greenhouse gas fluxes in agroecosystems

**Strengths:**

1. The paper tackles an important environmental and agricultural challenge: modeling carbon and nitrogen fluxes to support climate-change mitigation. The introduction provides solid context linking agriculture, greenhouse gas emissions, and the need for AI-ready benchmarks.

2. The authors define consistent data splits, metrics, and tasks for both simulated and observational datasets, facilitating fair model comparison.

3. Baseline results for six state-of-the-art time-series models (LSTM, TCN, Transformer variants) are carefully reported, providing useful empirical baselines for the community.

**Weaknesses:**

1. The major limitation is that AgroFlux primarily relies on synthetic simulations from process-based models (Ecosys and DayCent). Although the paper claims to integrate observational data, the latter is extremely limited in both spatial and temporal coverage (e.g., 11 flux-tower sites and one small-scale N₂O experiment).

2. The contribution of this work is primarily dataset engineering, and the idea of combining PBM simulations with flux-tower observations has been explored before (e.g., FLUXCOM, KGML-Ag). Thus, the work provides incremental improvements rather than a conceptual leap.

3. Because both the training and testing phases rely heavily on data derived from the same simulation pipelines, the evaluation risks validating models on artificial distributions, which may not transfer to real-world agroecosystem variability.

4. The claim that AgroFlux is the “first benchmark for agricultural flux prediction” overlooks existing efforts such as FLUXNET2015, GHG-Europe, and X-BASE/FLUXCOM, which already provide large-scale flux data. The novelty here lies mainly in the combination with simulations, not in benchmarking per se.

**Questions:**

1. Given that most of AgroFlux consists of model-simulated data, how do you ensure that the benchmark faithfully represents real-world agroecosystem dynamics rather than the internal biases of Ecosys and DayCent?

2. Can you justify why evaluating ML models primarily on simulated data provides meaningful insights for real-world deployment?

3. Have you assessed domain shift between simulated and observed distributions (e.g., feature statistics or covariate shift metrics)?

---

### Official Review · Reviewer_HE94 · 2025-10-31

**Soundness:** 1
**Presentation:** 2
**Contribution:** 2
**Rating:** 2
**Confidence:** 5

**Summary:**

This work introduces a dataset for carbon dioxide and nitrous oxide surface fluxes of soy- and maize agricultural lands in the midwest region of the US. The task is approached in a two-fold way: first by providing simulations from two process-based biogeochemical models over a gradient of environmental conditions, and then second, by assembling eddy covariance CO2 flux data from 11 sites alongside N2O flux data from six chamber experiments and colocating it with environmental covariates such as weather and soil conditions. The study then proceeds by training a range of sequence-based deep learning models such as LSTM-variants or Transformer-variants on the datasets and evaluating them in temporal and in spatial cross-validation for the simulated and the observed datasets. Finally, experiments are presented were models are first pretrained on the simulation data and then fine-tuned on the observations, which provides performance improvements in some cases.

**Strengths:**

1. The monitoring of greenhouse gas surface fluxes is highly relevant for verification of international climate agreements, and at the same time quite challenging due to gaps in our understanding of ecosystem processes and limited observations.
2. This work allows for a multi-task (NEE, GPP & N2O flux) training of models, with potential benefits across tasks.
3. The presented process-based model simulations allow for synthetic experiments to better illucidate the extrapolation capabilities of current models due to the limited observational network
4. The combination of simulation data and observations allows to study transfer learning from process-based models. This study presents first results that highlight potential benefits of such an approach.

**Weaknesses:**

Major comments:

1. A lot of the data has already been introduced in Liu et al 2024. What exactly is the contribution of this paper?
2. Insufficient baselines. FluxCom X-Base should be included, as should be an XGBoost trained on your dataset and the two process-based models Ecosys and Daycent.
3. Mediocre performance on the synthetic dataset. Why do you achieve only mediocre performance on a mere emulation task? Are your models too small or not properly tuned? Are you extrapolating in feature space?
4. You argue existing flux upscaling is not suitable partly because not enough Agri sites (L.97f.) - but then only use 11 agri sites.
5. This is also an argument itself: 11 sites is very few (!)
6. Some of the sites are very close-by. Thus you might overestimate spatial extrapolation skill. I suggest a spatially-blocked CV, as Fluxcom X-Base does.
7. Please cite Flux Data (and all other used datasets) correctly. For an example, see the license and citation for US-Bo1 here: https://ameriflux.lbl.gov/sites/siteinfo/US-Bo1
8. There should not be that great differences between the different models (especially for the emulation task in Table 1). How did you do the hyperparameter tuning?
9. Especially for N2O, where you are in a way more data-limited setting, it would be important to include simpler baselines
10. The title is too general, your benchmark is only valid for Corn & Soybean agricultural systems in 3 states of the US.
11. One important experiment lacks: You could use your simulated data for a synthetic observation system simulation experiment (OSSE), i.e. trying to assemble a training dataset whose distribution reflects the sparse distribution of observation stations, and to then test model extrapolation in such a synthetic set-up.
12. Experiment A.4 is potentially unfair: your task-specific models potentially have access to 4x the parameters.
13. It would be useful to extend A.4 also to the observational datasets, as there might be more benefits there from multi-task learning
14. Why does pretraining on Ecosys data (esp. Table 3, but also table 4) lead to much worse results than just training directly on observations (Table 2)? Also: to make this analysis easier. Best would be to add a "gain" column in Table 3 & 4 which shows the improvement over training from scratch.

Writing needs to be greatly improved:

15. Limitations is way to short and misses key aspects, for example problems due to missing covariates, limited transferability to arbitrary agricultural fields, … Especially N2O is a chamber measurement from the lab, and thus there is a gap between this data and real ecosystem response
16. It is not explained exactly which predictors are used for the observational data. And it is a bit fuzzy, which are used for the simulated data. Can you comment on the quality of spatial maps of these parameters in case you want to apply your model afterwards to produce maps?
17. The compared methods are not explained. It would be important to mention why these methods where chosen, and how they differ. To then draw potential conclusion on why a certain method outperforms over another.
18. Vast literature on flux upscaling using ML models not mentioned in introduction, and only 2 papers cited in related works. Also there are many works using sequence models. Cite them !
19. The abstract has poor language quality. First sentence has a missing word "are", and i believe you can't use "agroecosystem" without an article in this case. Disentangle the existing approaches and their respective challenges in line 18ff. Line 10 seems colloqiual, what is a “AI-ready benchmark dataset” or an “AI-empowered model”. Use scientific terms. Also be accurate, there are already benchmarks for evaluating data-driven flux models (e.g. you are citing Fluxcom-X).
20. Beyond this, language could also be improved throughout the rest of the manuscript.

Minor comments:

21. Please call it NEE and not CO_2 .
22. Specify which method is used to partition GPP (daytime or nighttime) in L.229.
24. I would prefer if R^2 colorbar would go from 0 to 1, and RMSE & MAE colorbars would start at 0 (e.g. in Fig. 3).
25. Add the performance gain for the pretrain-finetune strategies (e.g. as a color-code). So it becomes more clear the benefit
26. Add spread of your metrics across CV folds, i.e. uncertainty
27. In Fig.2 it looks like your models only learn to predict the mean seasonal cycle, and not much beyond. It could be interesting to, akin to X-Base, also explicitly disentengle this by reporting metrics for the MSC, the anomalies from MSC and the IAV next to the raw scores.
28. Fig. 1 the legend are overlapping the plotted data.

**Questions:**

See the Weaknesses above.

**Details Of Ethics Concerns:**

The used observational data has not been cited appropriately (as detailed in the data license - see https://ameriflux.lbl.gov/sites/siteinfo/US-Bo1#data-citation ).

---

### Official Review · Reviewer_XjEB · 2025-10-31

**Soundness:** 4
**Presentation:** 4
**Contribution:** 3
**Rating:** 6
**Confidence:** 4

**Summary:**

AgroFlux is a spatiotemporal benchmark suite designed to enhance machine learning–based prediction of agricultural greenhouse gas (GHG) fluxes, particularly carbon dioxide (CO₂) and nitrous oxide (N₂O). Recognizing the computational limits and biases of traditional process-based models (PBMs) like Ecosys and DayCent, AgroFlux integrates simulation outputs with real-world observational data from flux towers and controlled environments to provide standardized, reproducible datasets. It defines multiple prediction tasks, temporal and spatial extrapolation, simulation-to-observation transfer, and cross-domain learning, and evaluates several baseline and transfer learning models to establish reference performance. AgroFlux offers a consistent foundation and leaderboard for advancing robust, generalizable GHG flux prediction in agricultural systems.

**Strengths:**

1. Comprehensive integration of PBM simulation data and real-world observational datasets at daily granularity.
2. Covers a wide range of environmental and management variables across different sites and conditions.
3. Standardized prediction tasks and consistent evaluation metrics (R2, RMSE, MAE) enable fair, reproducible assessment.
4. Includes transfer learning benchmarks, pushing forward domain adaptation research.
5. Provides baseline performances on state-of-the-art sequential deep learning models (LSTM variants, TCN, Transformers).
6. Supports development of accurate, scalable AI-driven models to better understand and mitigate agroecosystem climate impacts.

**Weaknesses:**

1. The dataset may still be limited to certain regions or crop types, possibly restricting generalizability.
2. Complexity in data integration from multiple sources may pose application challenges.
3. Machine learning models’ performance could be sensitive to the high spatio-temporal variability of agricultural fluxes.
4. PBMs themselves have inherent biases that might propagate into benchmarks.
5. Only a data-driven approach might not be sufficient to capture vastly complex agricultural fluxes.
6. The benchmark did not incorporate all regression-based models, such as State Space models.
7. The paper does not benchmark the computational costs of the considered models.

**Questions:**

1. How do discrepancies between simulated (PBM) data and observational data affect benchmark robustness? Are there plans to mitigate PBM structural biases in the dataset?
2. How can the N2O flux prediction be benchmarked with higher quality? Is there a plan/algorithm/model for such a task?
3. Are there any results for State Space models?
4. Although N2O flux prediction performs very well based on MAE and RMSE, it performs lower than CO2 and GPP. The paper mentions that this is due to management practices. Additional results or evidence in support of this statement can help assess the paper.
5. Benchmarks of models on the Flux prediction datasets provide insights into models only based on the considered metrics, but not on the computational complexity and expense of the models. Presentations of such results help to assess the paper comprehensively.

---

### Official Review · Reviewer_sfFj · 2025-11-01

**Soundness:** 2
**Presentation:** 2
**Contribution:** 1
**Rating:** 0
**Confidence:** 4

**Summary:**

The paper presents a spatio-temporal benchmark for the prediction of geophysical variables. It evaluates several architectures (LSTMs, 1D-CNNs, Transformer-based models) across multiple tasks.

While this is valuable work for the biophysical / geoscience community, the impact for the machine learning community feels limited. The empirical results are somewhat inconclusive — depending on the task, different architectures “win”, with no clear take-home message on what matters model- or data-wise. In addition, the study would benefit from including a domain-specific baseline (e.g. a random forest on hand-crafted features), to better contextualize how domain experts approach these problems today and to quantify the real ML lift.

Overall, I do not see a strong fit for ICLR. However, I do see this as a good paper for a domain journal or a domain-specific workshop, where the target audience would likely appreciate its contributions more directly.

**Strengths:**

* Extensive empirical comparison of multiple deep learning architectures (LSTM, 1D-CNN, Transformer) across a diverse set of biogeophysical prediction tasks.
* The paper reflects substantial domain knowledge from the biogeosciences, both in how variables are selected/aggregated and how the tasks are framed.

**Weaknesses:**

* Limited relevance for the broader ML community outside biogeosciences. The paper does not clearly articulate which underlying modeling patterns or dataset properties are generalizable beyond this specific domain, or why this benchmark represents a unique opportunity for ML research at large.
* Results are largely inconclusive and there is no methodological innovation. The baseline comparisons do not yield a clear insight or takeaway that advances our understanding of model behavior or design principles.

**Questions:**

In terms of topical alignment:
* What underlying patterns are visible in the results that a ML- researcher from a different application domain could utilize for their research?
* What are the unique patterns in the data of this benchmark that makes this benchmark a unique opportunity for the ML community as a whole?

Technical Question
* How relieable are the simulated data for a benchmark? As the simulation follows presumably a mathematical model for the underlying process and a noise model, I see a clear threat of deep learning models approximating the underlying functions.

---

### Note · Authors · 2025-11-30

I have read and agree with the venue's withdrawal policy on behalf of myself and my co-authors.